# Factor Structure and Psychometric Properties for the PTSD Checklist of Chinese Adolescents in the Closed Period after the COVID-19 Outbreak

**DOI:** 10.3390/ijerph182212245

**Published:** 2021-11-22

**Authors:** Wei Chen, Rongfen Gao, Tao Yang

**Affiliations:** School of Psychology, Guizhou Normal University, Guiyang 550025, China; 19010250539@gznu.edu.cn (R.G.); 19010250522@gznu.edu.cn (T.Y.)

**Keywords:** adolescents, PTSD, COVID-19, confirmatory factor analysis, measurement invariance

## Abstract

After COVID-19 appeared in China in December 2019, the mental health of adolescents, as a vulnerable group in public health emergencies, was negatively affected by the epidemic and the unprecedented prevention and control measures. The purpose of this study was to investigate the factor structure and psychometric properties of the Posttraumatic Stress Disorder (PTSD) Checklist (PCL) among Chinese adolescents. A total of 915 participants completed the PTSD. Confirmatory factor analyses (CFAs) and multi-group CFAs were used to test the factor structure and psychometric properties of PTSD. The CFA results showed that five-factor PCL was the optimal fitting model with satisfactory reliability and validity; moreover, it was suggested that the properties of PCL were invariant across gender, PTSD and asymptomatic groups, early and late adolescents, as well as over time. In summary, PCL is applicable among Chinese adolescents and can be used for effective measurement of PTSD caused by epidemics and to conduct cross-group studies.

## 1. Introduction

Coronavirus disease (COVID-19) emerged in Wuhan, China, in December 2019 and grew into a pandemic by March 2020 [1,2,3]. As of 9 November 2021, there have been 126,782 COVID-19 patients and 5697 deaths related in China. Of these, 159 patients with COVID-19 and 2 deaths were in the Guizhou Province. At the time of the worst outbreak of the COVID-19, China played a metaphorical game of chess; all people were isolated at home and protested together. Early research [4,5] and recent findings [6,7] suggested that infectious disease epidemics and pandemics may be traumatic experiences for some people that may lead to post-traumatic stress disorder (PTSD) [3] and chronic psychological symptoms. Since the early days of the pandemic, public health experts have noted that the prevalence of PTSD is likely to increase in the general population [8]. Early data indicated an increase in the prevalence of PTSD and traumatic symptoms in the general population since the COVID-19 epidemic began [9,10,11]. The mental health of adolescents, as vulnerable groups in public health emergencies, was negatively affected by the COVID-19 outbreak and the unprecedented measures implemented to curb its spread [12,13]. Adolescents were at a high risk of multiple mental health problems and experienced post-traumatic stress disorder (PTSD) [14,15].

PTSD is a persistent and severe mental disorder that occurs after individuals are exposed to an unusually threatening and catastrophic event [16]. Among people who experience a traumatic injury, PTSD is one of the strongest factors associated with post-traumatic life quality and recovery, especially when compared to physically traumatized individuals without PTSD [17]. The PTSD Checklist (PCL) is one of instruments used to measurement the level of PTSD, comprising 17 items based on the fourth Diagnostic and Statistical Manual of Mental Disorder (DSM-IV) or 20 items based on Diagnostic and Statistical Manual of Mental Disorders, 5th Edition (DSM-V). Although previous studies have pointed out that the models based on DSM-V have certain evidence to support them, the symptom structure of these models is relatively dispersed, which may make the diagnosis of PTSD more extensive [16,18]. Therefore, it is of great significance to further investigate the structure of PTSD based on DSM-IV [16,19]. More importantly, the classification of symptom structures for PTSD in DSM-V is mainly influenced by the four-dimensional model of emotional numbness by King et al. (1998) and the four-dimensional model of mental distress by Simms et al. (2002) [20]. It can be seen that further investigation of the structural model of DSM-IV can provide reference for improving of the structure of PTSD symptoms in DSM-5.

For confirmation, studies have explored the factor structure of PTSD among Chinese adolescents [16,21]. However, the factor structure identified in these studies is not necessarily applicable to the assessment of trauma symptoms in adolescents under the COVID-19 context. Similar to Severe Acute Respiratory Syndromes (SARS), COVID-19 is another shocking epidemic event, but the latter has spread more widely, leading to more hospitalizations and deaths worldwide. In addition, COVID-19 continues to spread rapidly around the world, affecting more people every day in various ways (e.g., economic losses, unemployment, difficulties in obtaining important materials, increased social isolation, uncertainties about the future); therefore, the impact of the COVID-19 pandemic on mental health will be more extensive and possibly more far-reaching than the SARS epidemic [7,22]. For these reasons, if the measurements used in a context other than the one for which it they were developed, then they are likely to perform differently. It is critical to evaluate the performance of these tools in various application environments where they are used [17,23,24]. Therefore, this research aimed to explore the structure of PTSD symptoms based on DSM-IV under the COVID-19 outbreak.

Additionally, during the COVID-19 outbreak, home isolation has emerged as one of the main forms of protection; however, the prevalence of PTSD is particularly high in self-isolated populations [11,25] and previous studies have shown that increasing social support helps to reduce PTSD [3,26]. Compared with resilient responders, patients with PTSD have significantly higher levels of dysfunction within 2 years after injury [27]. This effect was demonstrated to be significant for 6 years after the injury, with residual dysfunction even after symptom relief [17,27]. In addition to the external factors mentioned above, gender difference is also one of the demographic factors that scholars pay the most attention to. Numerous studies have shown that there is a lower incidence of PTSD in males than females [3,7,22,28,29,30,31]. It is true that previous studies have obtained many intentional conclusions, but it is not clear whether the explanation of these conclusions is valid. Therefore, it is necessary to test the invariance of PCL for assessing PTSD. Additionally, according to the age classification of the World Health Organization [32], adolescents are 10–19 years old: early adolescents are 10–14 years old and late adolescents are 15–19 years old. On the basis of previous research theories, the current study also divided the participants into groups to explore the measurement invariance of PCL in early and late adolescents and to provide a strong basis for the comparative study of PTSD in early and late adolescents.

### The Current Study

The aims of the current study were to examine the factor structure and psychometric properties of the PTSD in mainland Chinese adolescents. First, we discuss the optimal factor structure of PCL through confirmatory factor analyses (CFAs). In this process, the models we tested include: The single-factor model (M1) [33]; the two-factor model (M2) [34]; the three-factor model (M3) [35]; the four-factor emotional numbing PTSD model (M4a) [36]; the four-factor dysphoria PTSD model (M4b) [37]; at the same time, we also tested the second-order factor structure of the emotional numbing PTSD model (M5a) and the dysphoria PTSD model (M5b), that is, adding a second-order factor to the M4a,b models; and the dysphoric arousal model (M6) [38]. As shown in previous studies, we assumed that the five-factor model was the optimal model.

Furthermore, the reliability of the optimal factor structure based on CFAs was also examined, including the internal consistency coefficient (Cronbach’s α) of potential factors [39], the scalability of dimensions (Loevinger’s H) [40], and consistency between items and dimensions (Hj-min) [40]. In addition, the convergent and discriminant validities of five-factor PCL were determined through the examination of a correlation matrix; the elements of this matrix were the correlation coefficients between items and rest scores [40].

Finally, we tested the measurement invariance (MI) of PCL. MI is a statistical property that determined whether the items used in a questionnaire had the same meaning for different groups of participants. If MI cannot be established, the mean difference in observed values between groups cannot be directly explained [41]. This makes it difficult to draw conclusions about traditionally observed mean differences in various aspects, including cross-sectional study (e.g., sex) and longitudinal study (mainly referring to different time groups) [42,43]. Based on the existing theoretical basis, we explored the MI of PCL across gender, symptomatic and asymptomatic groups, early and late adolescents; additionally, the longitudinal measurement invariance was also tested.

## 2. Methods

### 2.1. Participants

In September 2020, cluster sampling was used and the participants in current study were from 6 schools in Guizhou, China. A total of 915 adolescents participated in the study. Their ages ranged from 11 to 18 years (age was not reported for 3 of the participants, who were coded as missing) and the mean age was 14.19 years (SD = 1.29); 476 (52%) participants were boys and 439 (48%) were girls. Three months later, students from 2 of the 6 schools were tested for the second time, and 300 valid questionnaires were collected. An independent sample *t*-test showed that there was no significant difference in the total score of PCL at the first time point (T1) between the participants and dropouts at the second time point (T2) (t = −0.03, *p* = 0.97), indicating that the sample loss at T2 was random.

### 2.2. Instruments

Posttraumatic Stress Disorder (PTSD) Checklist (PCL). The PCL was developed based on DSM-IV with 17 items [44]. As it has previously been translated by domestic scholars [45,46], it was not translated in present study. For each item, participants were asked to indicate how much they had been disturbed by each symptom in the past month on a 5-point Likert scale ranging from 1 (“never”) to 5 (“extremely”). The PCL total score (the total score for the 17 items) ranged from 17 to 85. A higher score indicates a higher level of PTSD, and a score of 38 or higher is considered likely to indicate PTSD [47,48]. The internal consistency coefficient of the scale was 0.92.

### 2.3. Procedure

We obtained consent from the participants’ guardian and school leaders before the study began; indeed, after we told principals the purpose of our study, strictly abided by the principle of confidentiality, and helped the school screen the students’ mental health problems, we were unanimously recognized by principals. Subsequently, the teachers of each class informed the parents of the survey through the home–school cooperative Wechat group. The participants were required to sign an informed consent form before they completed the paper-and-pencil questionnaire in the classroom and the researcher collected the questionnaires uniformly, which took about 15–20 min. After the questionnaires were collected, EpiData 3.1 (The EpiData Association, Denmark, Europe) was used to build the database, and two researchers entered the data independently. The current study was conducted in line with the Helsinki Declaration of Ethical Principles and was approved by the Committee of the School of Psychology of Guizhou Normal University.

### 2.4. Analytical Plan

First, descriptive statistics of the whole scale were performed by STATA/SE 13.1 (StataCorp LLC, College Station, TX, USA) [49]. Second, a series of CFAs were conducted through MPLUS 8.3 (The National Institute on Alcohol Abuse and Alcoholism, National Institutes of Health, Los Angeles, CA, USA) [50] to test and compare the PTSD model mentioned above. The skewness and kurtosis values of some items were out of the acceptable range (skewness ± 3.00, kurtosis ± 8.00), indicating that the sample distribution was non-normal [51]; therefore, the mean adjusted maximum likelihood estimator (MLM) estimation method was used for data processing in this study [52].

In this study, the following indicators were used to evaluate the model fitting: root mean square error of approximation (RSMEA); standardized root mean square residual (SRMR); the Tucker–Lewis index (TLI); comparative fit index (CFI); Akaike information criterion (AIC), and Bayesian information criterion (BIC). Generally speaking, for CFI and TLI, values greater than 0.90 and 0.95 are considered to reflect acceptable and optimal fit to the data, respectively. For RMSEA, values less than 0.08 and 0.06 were regarded as reasonable and best fitting indices for the data, respectively [53] An SRMR value of less than 0.08 indicates a good fit of the model [54]. For non-nested models, the BIC difference is compared to judge the model’s advantages and disadvantages. If the difference of BIC between two non-nested models is greater than 10, it indicates that there is a large difference between the two models. At this time, the model with a smaller BIC value should be selected as the optimal model [55].

In accordance with previous guidelines, the αs were interpreted as follows: α < 0.60 (unacceptable), α = 0.60–0.65 (undesirable), α = 0.65–0.70 (minimally acceptable), α = 0.70–0.80 (respectable), α = 0.80–0.90 (excellent), and α > 0.90 (excessive consistency) [56]. Moreover, the Loevinger’s H coefficient of the subscales of >0.30 suggests satisfactory scalability, and an Hj-min of >0.30 indicates that items might be consistent with the subscale [40]. Meanwhile, Perrot et al. (2018) point out that elements on the diagonal greater than 0.40 suggest there is convergent validity, while off-diagonal values greater than the values on the diagonal demonstrate a lack of divergent validity.

Subsequently, the entire sample was used to examine the MI across gender, the MI across symptomatic and asymptomatic groups, early adolescents and late adolescents, as well as the longitudinal MI. The diagnostic cutoff point of 38 points was used as the cutoff point for those with and without symptoms of PTSD. In this study, 761 (83.20%) participants scored less than 38, and 154 (16.80%) participants scored greater than or equal to 38. Additionally, we analyzed the frequency of age and found that there were too few participants of several ages to be representative, so we considered removing them (12 participants were excluded). Then, the remaining 903 participants were divided into two groups: early adolescence (N = 436) and late adolescence (N = 467). MI was established through a multi-group CFA (MGCFA) common stage framework [57], including: (a) configural invariance (e.g., no parameters are set to be equal across groups), (b) weak invariance (e.g., factor loadings are allowed to be equal across groups), (c) scalar invariance (e.g., the factor loadings and intercepts are allowed to be equal across groups), and (d) strict invariance (e.g., the factor loadings, intercepts, and unique factor variances are allowed to be equal across groups) [57]. A more rigorous model test was performed only if the previous measurement model was satisfied. Specifically, before the MI test, we carried out a single test (e.g., male and female groups; participants with PTSD symptoms and those without PTSD symptoms; early adolescents and late adolescents, as well as those who were measured twice). For nested models, the equivalent model was considered acceptable when ∆CFI ≤ 0.01 and ∆TLI ≤ 0.01 [41].

## 3. Results

### 3.1. Descriptive Statistics

Before exploring the factor structure, the descriptive statistical analysis of the total sample was prepared, as presented in Table 1. As can be seen, there was a reasonable amount of dispersion in the study variables. The mean PTSD mean score was 27.44 and the SD was 11.24. The reexperiencing (R) subscale had a mean of 7.90 with a SD of 3.84; Avoidance (A) had a mean value of 3.06 and a SD of 1.76. Numbing (N) had a mean of 7.47 with a SD of 3.34. Dysphoric arousal (DA) evidenced a mean of 5.14 with a SD of 2.71. The anxious arousal (AA) in PTSD had a mean of 3.86 and an SD of 2.30.

### 3.2. Factor Structure of the PTSD

As indicated in Table 2, the 17-item model fit indices for the single-factor model (M1), the two-factor model (M2), the three-factor model (M3) were poor; the model fitting indices (CFIs and TLIs) were not up to standard. All the other models (M4a, M4b, M5a, M5b, M6) reached the fitting standard; the fitting indices of M6 were better than that of other models (MLM χ^2^ = 375.76, df = 109, CFI = 0.97, TLI = 0.96, RMSEA = 0.05, SRMR = 0.03, AIC = 34,680.28, BIC = 34,974.24), with the smallest values of AIC and BIC, and the ratio of ΔBIC of the non-nested models was greater than 10. The factor loading of the five-factor Dysphoric Arousal model detail in Figure 1, and follow-up studies were based on this model.

At the same time, we also examined the reliability and validity of the latent factor structure. Cronbach’s alpha was greater than 0.70, and Loevinger’s H and Hj-min were greater than 0.30. Additionally, the correlations between an item and the rest score of its dimension was greater than 0.40, and the correlation between the item and the dimension that was not part of itself was smaller than that of the dimension to which it belonged (see Table 3).

### 3.3. Measurement Invariance

MI across gender of the five-factor PCL was examined in the total sample. First, model fits were examined for male and female participants respectively, and all model fitting indices were adequate. For all invariance testing, the model fitting indices were satisfactory (e.g., CFI, TLI > 0.90, and RMSEA, SRMR < 0.08, ΔCFI and ΔTLI < 0.01). Overall, results suggested that the PCL scores were invariant across gender of adolescents (see Table 4).

MI across participants with PTSD symptoms and those without PTSD symptoms of the five-factor PCL was examined. First, model fits were examined separately for with or without PTSD symptoms reports, and satisfactory model fitting results were obtained. For the three invariance models, the fit indices were adequate (e.g., CFI, TLI > 0.90, and RMSEA, SRMR < 0.08, ΔCFI and ΔTLI < 0.01). All in all, results suggested that the PCL scores were invariant across with or without PTSD symptoms of adolescents (see Table 4).

MI across early adolescents and late adolescents of the five-factor PCL was examined. Firstly, model fits were examined separately for early adolescents and late adolescents, satisfactory model fitting results are obtained. For all the invariance models, the fit indices were adequate (e.g., CFI, TLI > 0.90, and RMSEA, SRMR < 0.08, ΔCFI and ΔTLI < 0.01). In a word, findings demonstrated that the PCL scores were invariant across early adolescents and late adolescents of adolescents (see Table 4).

MI across different time group of the five-factor PCL was examined with data from repeated measurements. Indeed, model fits were examined respectively for the participants participated in this study for the T1 and who participated in this study for the T2, and the results indicated that the model fitting was satisfactory (e.g., CFI, TLI > 0.90, and RMSEA, SRMR < 0.08, ΔCFI and ΔTLI < 0.01). In conclusion, the results revealed that the PCL scores were invariant across time for adolescents, as detailed in Table 4. 

## 4. Discussion

This is the first study to examine the factor structure and psychometric properties of the PCL in a sample of adolescents from mainland China under the COVID-19 outbreak. Through CFAs, we found that the five-factor model of PCL was the best fit. The reliability and validity of this factor model were proved to be satisfactory. Most importantly, the MI results indicated that the optimal five-factor PCL had strict invariance across gender and early and late adolescents and strong invariance across the with or without PTSD symptom group and longitudinal MI of adolescents.

As expected, the five-factor model had the optimal fit. First, in terms of model fitting, the three models M1–3 failed to meet the fitting standards. Although the four models M4a–M5b reached satisfactory fitting standards (CFI, TLI > 0.90, RMSEA, SRMR < 0.80), the AIC and BIC values of these models were not the smallest. Relatively speaking, the five-factor model was an optimal fit to the data (CFI, TLI > 0.95, RMSEA < 0.6, SRMR < 0.80) [53], which is consistent with the results of previous studies [16,38,58]. Second, from a theoretical point of view, the five-factor theory is also more convincing. In the emotional numbness model [36], D1–D3 symptoms are placed in the hyperarousal factor, which is different from the dysphoria model [37] where D1–D3 symptoms were placed in the dysphoria factor. However, it can be argued that D1–D3 symptoms are conceptually distinct in hyperarousal and anxiety disorders. Importantly, the five-factor model has the advantage of combining the mixed findings that typically occur in modern PTSD CFA research, with some finding support for the emotional numbness model and others finding support for the dysphoria model [38].

The Cronbach alpha for the PTSD was high both on the total scale and its subscale, all of them greater than 0.70; there was good-to-excellent reliability [56], which supported the results of previous studies [38]. Loevinger’s H coefficients for the five subscales were >0.30, which demonstrated good scalability. Meanwhile, the Hj-min coefficients were greater than 0.30, indicating that there was good consistency between the items and the dimensions. Additionally, the elements in the correlation matrix suggested that there was good convergent and discriminant validity [40].

Since few previous studies have directly and comprehensively tested the measurement invariance of PCL [59], further research on this topic was needed to determine the feasibility of our findings in different groups, such as gender, with or without PTSD symptom, early and late adolescents, as well as over time. The analysis of measurement invariance also provided some useful insights into the measurement characteristics of PTSD. Strict invariance of PTSD across gender, early and late adolescents were observed for all five sub-scales; the fitting indices of CFI and TLI were more than 0.90, RMSEA and SRMR were less than 0.08 for all models, and the comparison results of all nested models showed that ΔCFI and ΔTLI were less than 0.01, indicating that the scores of these five sub-scales could be interpreted in the same way for both male and female, early and late adolescents, and the difference in test performance among different groups was due to the difference of potential variables, not the difference caused by artificial factors [60]. Similarly, we also found that PCL had strong invariance between PTSD symptoms and asymptomatic individuals, as well as at different time points. It can be seen that group variables did not affect the effectiveness of the questionnaire in measuring individuals’ PTSD. The above measurement invariance conclusions showed that in practice and research, we can compare the differences between the symptomatic and asymptomatic groups, and the specific changes in PTSD symptoms of the same individual over time. In conclusion, the current findings showed that PCL was suitable for Chinese adolescents, provided a solid theoretical basis guidance for empirical research and practice, and should convince researchers and educators to use this tool to measure and explain PTSD.

## 5. Conclusions

COVID-19 has caused great harm to people’s physical and mental health, and of those affected have PTSD. The current research results indicated that PCL can be used effectively to evaluate the PTSD caused by COVID-19 in Chinese adolescents. Specifically, PCL has a satisfactory factor structure, and the five-factor dysphoric arousal model is stable, with satisfactory reliability and validity. In addition, the five-factor model achieved strict invariance between gender, early and late adolescents, and strong invariance between PTSD groups and asymptomatic groups, as well as longitudinal invariance. The PCL has good psychometric properties in Chinese adolescents under the COVID-19 outbreak.

## Figures and Tables

**Figure 1 ijerph-18-12245-f001:**
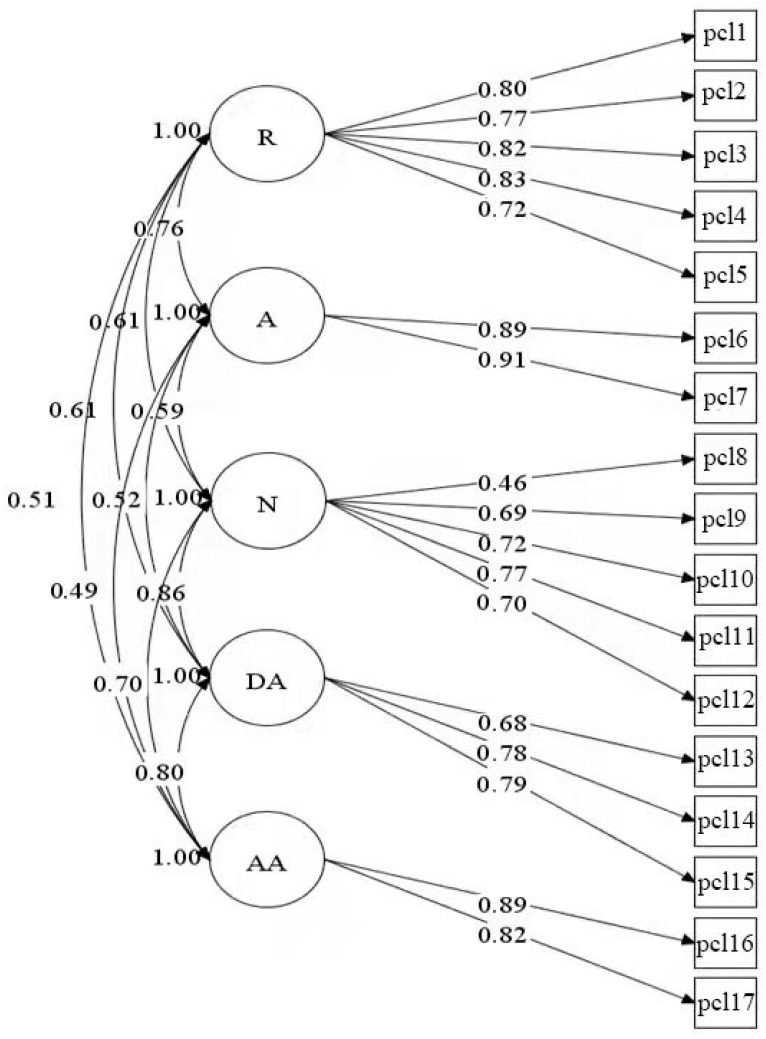
Factor loading of the five-factor PCL. R = reexperiencing; A = avoidance; N = numbing; DA = dysphoric arousal; AA = anxious arousal.

**Table 1 ijerph-18-12245-t001:** Descriptive statistics of PTSD Checklist (N = 915).

Items	*Min*	*Max*	*M*	*SD*	M6	Skewness	Kurtosis
pcl1. Intrusive thoughts	1	5	1.82	1.01	R_*M* ± *SD*_ = 7.90 ± 3.84	1.33	4.37
pcl2. Nightmares	1	5	1.36	0.78		2.69	10.84
pcl3. Reliving trauma	1	5	1.65	1.02		1.63	5.03
pcl4. Emotional cue reactivity	1	5	1.75	1.04		1.51	4.80
pcl5. Physiological cue reactivity	1	5	1.32	0.73		2.87	12.04
pcl6. Avoidance of thoughts	1	5	1.54	0.93	A_*M* ± *SD*_ = 3.06 ± 1.76	1.90	6.21
pcl7. Avoidance of reminders	1	5	1.52	0.92		1.99	6.64
pcl8. Trauma-related amnesia	1	5	1.47	0.80	N_*M* ± *SD*_ = 7.47 ± 3.34	1.88	6.64
pcl9. Loss of interest	1	5	1.40	0.81		2.28	8.18
pcl10. Feeling detached	1	5	1.55	0.91		1.85	6.14
pcl11. Feeling numb	1	5	1.56	1.01		1.98	6.24
pcl12. Hopelessness	1	5	1.49	0.94		2.13	7.07
pcl13. Difficulty sleeping	1	5	1.62	1.05	DA_*M* ± *SD*_ = 5.14 ± 2.71	1.84	5.65
pcl14. Irritable/angry	1	5	1.76	1.10		1.48	4.33
pcl15. Difficulty concentrating	1	5	1.77	1.07		1.41	4.27
pcl16. Overly alert	1	5	1.93	1.25	AA_*M* ± *SD*_ = 3.86 ± 2.30	1.22	3.33
pcl17. Easily startled	1	5	1.93	1.22		1.21	3.41
PCL	17	85	27.44	11.24	-	-	-

M6 = Dysphoric Arousal Model; PCL = Posttraumatic Stress Disorder (PTSD) Checklist; R = reexperiencing; A = avoidance; N = numbing; DA = dysphoric arousal; AA = anxious arousal.

**Table 2 ijerph-18-12245-t002:** Confirmatory factor analyses model fit statistics of the PCL.

Model	MLM χ^2^	df	CFI	TLI	RMSEA	SRMR	AIC	BIC	ΔBIC
M1	1286.48	119	0.73	0.69	0.10	0.09	36,823.23	37,069.00	
M2	606.02	118	0.89	0.87	0.07	0.05	35,468.54	35,719.13	−1349.87
M3	698.10	116	0.87	0.84	0.07	0.06	35,661.65	35,921.87	202.74
M4a	575.68	113	0.95	0.94	0.07	0.04	34,872.21	35,146.89	−774.98
M4b	501.19	113	0.96	0.95	0.06	0.04	34,797.72	35,072.40	−74.49
M5a	777.02	115	0.92	0.91	0.08	0.06	35,069.55	35,334.59	262.19
M5b	701.70	115	0.93	0.92	0.08	0.06	34,994.23	35,259.27	−75.32
M6	375.76	109	0.97	0.96	0.05	0.03	34,680.28	34,974.24	−285.03

M1 = Unidimensional; M2 = Two-factor model; M3 = DSM-IV; M4a = emotional numbing model; M4b = dysphoria model; M5a = second-order emotional numbing model; M5b = second-order dysphoria model; M6 = dysphoric arousal model.

**Table 3 ijerph-18-12245-t003:** Reliability and validity statistics of five-factor PCL.

Items	R	A	N	DA	AA
α	0.89	0.90	0.80	0.79	0.84
H	0.53	0.55	0.50	0.56	0.52
Hj-min	0.64	0.82	0.34	0.54	0.74
pcl1	0.75	0.58	0.42	0.39	0.35
pcl2	0.73	0.55	0.45	0.43	0.34
pcl3	0.76	0.56	0.47	0.46	0.41
pcl4	0.77	0.62	0.42	0.44	0.40
pcl5	0.67	0.52	0.47	0.46	0.33
pcl6	0.64	0.81	0.47	0.40	0.40
pcl7	0.65	0.81	0.51	0.44	0.42
pcl8	0.31	0.34	0.40	0.35	0.29
pcl9	0.40	0.40	0.61	0.53	0.43
pcl10	0.43	0.44	0.65	0.51	0.46
pcl11	0.45	0.40	0.65	0.62	0.50
pcl12	0.36	0.34	0.60	0.58	0.47
pcl13	0.40	0.35	0.56	0.57	0.51
pcl14	0.46	0.40	0.58	0.67	0.58
pcl15	0.45	0.36	0.62	0.67	0.57
pcl16	0.43	0.41	0.56	0.64	0.73
pcl17	0.39	0.40	0.52	0.59	0.73

R = reexperiencing; A = avoidance; N = numbing; DA = dysphoric arousal; AA = anxious arousal.

**Table 4 ijerph-18-12245-t004:** Measurement invariance model fit statistics for the PCL.

Model	MLM χ^2^	df	CFI	TLI	RMSEA	SRMR	AIC	BIC	ΔCFI	ΔTLI
Across Gender										
Female	185.47	109	0.97	0.96	0.04	0.04	16,340.87	16,590.03		
Male	138.87	109	0.99	0.98	0.02	0.03	18,326.45	18,580.54		
Configural	597.25	218	0.96	0.95	0.06	0.04	34,667.33	35,255.24		
Weak	615.73	230	0.96	0.95	0.06	0.04	34,661.80	35,191.88	0.00	0.00
Scalar	632.80	242	0.96	0.95	0.04	0.04	34,654.87	35,127.12	0.00	0.00
Strict	688.76	259	0.95	0.95	0.06	0.05	34,676.83	35,067.16	−0.01	0.00
Across with or without PTSD Symptom										
With PTSD	119.27	109	0.98	0.98	0.03	0.06	8104.46	8289.72		
Without PTSD	206.46	109	0.95	0.94	0.03	0.05	22,656.89	22,939.76		
Configural	465.62	218	0.94	0.93	0.05	0.05	30,696.44	31,284.34		
Weak	513.16	230	0.93	0.92	0.05	0.05	30,719.98	31,250.06	−0.01	−0.01
Scalar	582.49	242	0.92	0.91	0.06	0.06	30,765.31	31,237.56	−0.01	−0.01
Across early adolescents and late adolescents										
early adolescents	244.62	109	0.97	0.96	0.05	0.03	16,935.14	17,183.88		
late adolescents	288.62	109	0.96	0.95	0.06	0.04	17,186.74	17,439.67		
Configural	533.24	218	0.96	0.96	0.06	0.04	34,121.88	34,708.18		
Weak	559.35	230	0.96	0.96	0.06	0.04	34,123.99	34,652.62	0.00	0.00
Scalar	568.81	242	0.96	0.96	0.06	0.04	34,109.45	34,580.41	0.00	0.00
Strict	615.46	259	0.96	0.96	0.06	0.04	34,122.10	34,511.37	0.00	0.00
Across Time										
T1	342.25	109	0.94	0.92	0.08	0.05	9816.92	10,042.65		
T2	280.46	109	0.94	0.92	0.07	0.05	11,353.39	11,579.32		
Configural	1034.45	465	0.92	0.90	0.06	0.05	20,775.30	21,382.72		
Weak	1059.36	477	0.92	0.90	0.06	0.05	20,776.21	21,339.18	0.00	0.00
Scalar	1090.63	489	0.91	0.90	0.06	0.05	20,783.48	21,302.01	−0.01	0.00

## Data Availability

This paper provides the data in this study.

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
