# Peer review of "Factor Structure and Psychometric Properties for the PTSD Checklist of Chinese Adolescents in the Closed Period after the COVID-19 Outbreak"

_ijerph, 2021, doi:10.3390/ijerph182212245_

Round 1

Reviewer 1 Report

As a large-scale study to examine the factor structure and psychometric properties of the PTSD among Chinese adolescents in the context of the COVID 19 pandemic, the results of the study were acceptable and interesting. There are some corrections for the reader's understanding, so it would be good to revise and publish the following items.

2.2. Instruments

PCL is described as 'For each item, participants were asked to indicate how much they had been disturbed by each symptom in the past month on a 5-point Likert scale ranging from 0 (“never”) to 5 (“extremely”). The PCL total score (the total score for the 17 items) ranges from 17 to 85', but if 'never' is 0, it is a 6-point scale, and it is also not correct that the range of scores is 17 to 85 and in Table 1, each min value is 1. Please check and correct.

2.3. Procedure

Since participants are between 11 and 18 years of age, minors are included. Please describe in more detail the process of obtaining permission from the school and guardians. And out of all 6 schools, how many students did not participate because their parents did not consent to participate in the study?

2.4. Analytic Plan

Usually, α is based on less than 0.05, but what is the rationale for stating that it is based on less than 0.6?

Since the age range of the subjects is large, 11 to 18 years old, and the way they cope with the social environment will be different, I would recommend looking at PTSD symptoms by dividing the age into early adolescents (11-13 years old children) and late adolescents (14-18 years old).

3.1. Descriptive statistics

Please, enter the mean and standard deviation of the sub-areas (R,A,N,DA,AA) and the entire tool in Table 1 as described in '3.1. Descriptive statistics'. How the 17 items of the PTSD Checklist are grouped into 5 sub-areas (R,A,N,DA,AA) should be identified in an easy-to-understand manner for the reader. By checking the 'Table 3. Reliability and validity statistics of five-factor PCL', it was possible to know which sub-factor each item was grouped into. It would be nice to be able to check the referenced tables as Tables 1 and 3.

'The anxious 184 arousal (AA) of the PTAD had a mean of 3.86 and an SD of 2.30.' Is PTAD a typo for PTSD in this sentence?

It is written as 'Overall, results suggested that the PCL scores were invariant across gender of adolescents (see Table 5)', but there is no table 5 in the text, so please check and re-enter it.

Please describe the guidelines for validating the model fit indices below Table 4.

4. Discussion

'The αs for the PTSD was high both total scale and its subscale, ~' In this sentence, if you write Cronbach alpha instead of α, it will be easier for readers to recognize.

Author Response

Response to Reviewer 1 Comments

Dear Reviewer,

Thank you for your valuable comments! In response to your comments, we have made a detailed reply. Please check its.

Reviewer #1:

Comments and Suggestions for Authors

As a large-scale study to examine the factor structure and psychometric properties of the PTSD amongChinese adolescents in the context of the COVID 19 pandemic, the results of the study wereacceptable and interesting. There are some corrections for the reader’s understanding, so it would be good to revise and publish the following items.

Point 1: 2.2. Instruments

PCL is described as “For each item, participants were asked to indicate how much they had been disturbed by each symptom in the past month on a 5-point Likert scale ranging from 0 (“never”) to 5 (“extremely”). The PCL total score (the total score for the 17 items) ranges from 17 to 85”, but if “never”is 0, it is a 6-point scale, and it is also not correct that the range of scores is 17 to 85 and in Table 1, each min value is 1. Please check and correct.

Re: Thanks for reviewers’ comments, we have changed 0 (“never”) to 1 (“never”). See the blue font on line 133 for details.

Point 2: 2.3. Procedure

Since participants are between 11 and 18 years of age, minors are included. Please describe in more detail the process of obtaining permission from the school and guardians. And out of all 6 schools, how many students did not participate because their parents did not consent to participate in the study?

Re: Thank you for your comments. a) We have described in detail the process of obtaining permission from the school and guardian. For details, see the blue font in lines 139-143, b) During the research, we firstly informed the teachers and guardians, carried out the research with their consent, and then obtained the specific number of participants. Unfortunately, there is no prior statistics on how many parents disagree with their children's participation in this study.

Point 3: 2.4. Analytic Plan

Usually, α is based on less than 0.05, but what is the rationale for stating that it is based on less than 0.6?

Re: Thanks for your comments, if I didn't understand wrong, In the expert's question, I saw what you said “α Is based on less than 0.05”, I wonder if you want to say that it is less than 0.5, and whether it is too small. At present, the journals we have consulted indicate that the alpha standard is more than 0.6 (Barker, 1994; Peterson, 1994), or a more stringent standard. Finally, we refer to the standard of 0.6 (Peterson, 1994). (e.g., Barker, C., Pistrang, N., & Elliot, R. (1994). Research methods in clinical and counselling psychology. Chichester: John Wiley & Sons). If necessary, I would also like to receive more detailed guidance from experts on this issue.

Point 4: Since the age range of the subjects is large, 11 to 18 years old, and the way they cope with the social environment will be different, I would recommend looking at PTSD symptoms by dividing the age into early adolescents (11-13 years old children) and late adolescents (14-18 years old).

Re: Thanks for your comments. According to expert suggestion, unfortunately, we did not find the literature source of the criteria for the division of adolescents pointed out in the expert's question. In the World Health Organizationage classification (World Health Organization, 2017), the age of adolescents is 10-19 years. In addition, adolescents are divided into younger adolescents (10-14 years) and older adolescents (15-19 years). First, we analyzed the frequency of age and found that there were too few participants of several ages to be representative, so we considered deleting them. Then, the remaining 903 participants were divided into two groups: early adolescence (N = 436) and late adolescence (N = 467), and relevant studies were carried out. The distribution of subjects is shown in Table 1. The contents of corresponding parts in the text have been modified and supplemented, for details see lines 82-87, 183-186, 251-256, table 4, blue font of the body.

Table 1. Descriptive statistics

age

count

grouping

11

3

delete

12

142

early adolescents (N=436)

13

118

14

176

15

374

late adolescents (N=467)

16

93

17

2

delete

18

4

delete

missing

3

-

total

915

903

Generally speaking, for CFI and TLI, values greater than 0.90 and 0.95 are considered to reflect acceptable and optimal fit to the data, respectively; For RMSEA, values less than 0.08 and 0.06 were regarded as reasonable and best fitting indexes with data, respectively (Marsh et al., 1999). where SRMR value less than 0.08 indicates a good fit of the model (Hu & Bentler, 1999 ). For non-nested models, BIC difference is compared to judge the model's advantages and disadvantages. If the difference of BIC between two non-nested models is greater than 10, it indicates that there is a large difference between the two models. At this time, the model with a smaller BIC value should be selected as the optimal model (Raftery,1995). Table 2 and table 3 showed the CFAs results of PCL in early and late adolescents. Comprehensively, we believed that the 5-factor model was the best fitting model in both early and late adolescents. It can be seen that there was the same factor structure in early adolescents and late adolescents.

Table 2. Factor Structure of PTSD among early adolescents(N=436)

Model

MLMχ2

df

CFI

TLI

RMSEA

SRMR

AIC

BIC

M1

711.23

119

0.74

0.70

0.11

0.09

18133.86

17925.90

M2

358.77

118

0.89

0.88

0.07

0.05

17304.51

17516.55

M3

412.60

116

0.87

0.85

0.08

0.07

17399.74

17619.94

M4a

348.84

113

0.94

0.93

0.07

0.04

17031.36

17263.78

M4b

285.58

113

0.96

0.95

0.06

0.04

16968.09

17200.52

M5a

435.65

115

0.92

0.91

0.08

0.06

17114.17

17338.44

M5b

371.81

115

0.94

0.93

0.07

0.05

17050.33

17274.60

M6

244.62

109

0.97

0.96

0.05

0.03

16935.14

17183.88

Table 3. Factor Analysis of PTSD among late adolescents (N=467)

Model

MLMχ2

df

CFI

TLI

RMSEA

SRMR

AIC

BIC

M1

678.01

119

0.74

0.70

0.10

0.10

18296.47

18507.93

M2

322.73

118

0.90

0.89

0.06

0.06

17540.00

17755.61

M3

384.55

116

0.87

0.85

0.07

0.07

17684.66

17908.56

M4a

367.74

113

0.94

0.93

0.07

0.04

17257.86

17494.20

M4b

375.12

113

0.94

0.93

0.07

0.05

17265.25

17501.59

M5a

507.54

115

0.91

0.90

0.09

0.07

17393.66

17621.71

M5b

510.12

115

0.91

0.90

0.09

0.07

17396.25

17624.29

M6

288.62

109

0.96

0.95

0.06

0.04

17186.74

17439.67

Based on the above results, we tested the measurement equivalence of PCL cross early adolescents and late adolescents. Multi group confirmatory factor analysis showed that PCL had strict measurement equivalence between early adolescents and late adolescents. See Table 4 for details.

Table 4. Measurement invariance model fit statistics for the PCL.

Model

MLMχ2

df

CFI

TLI

RMSEA

SRMR

AIC

BIC

ΔCFI

ΔTLI

Across early adolescents and late adolescents

early adolescents

244.62

109

0.97

0.96

0.05

0.03

16935.14

17183.88

late adolescents

288.62

109

0.96

0.95

0.06

0.04

17186.74

17439.67

Configural

533.24

218

0.96

0.96

0.06

0.04

34121.88

34708.18

Weak

559.35

230

0.96

0.96

0.06

0.04

34123.99

34652.62

0.00

0.00

Scalar

568.81

242

0.96

0.96

0.06

0.04

34109.45

34580.41

0.00

0.00

Strict

615.46

259

0.96

0.96

0.06

0.04

34122.10

34511.37

0.00

0.00

Point 5: 3.1. Descriptive statistics

Please, enter the mean and standard deviation of the sub-areas (R, A, N, DA, AA) and the entire tool in Table 1 as described in “3.1. Descriptive statistics”. How the 17 items of the PTSD Checklist are grouped into 5 sub-areas (R, A, N, DA, AA) should be identified in an easy-to-understand manner for the reader. By checking the“Table 3. Reliability and validity statistics of five-factor PCL”, it was possible to know which sub-factor each item was grouped into. It would be nice to be able to check the referenced tables as Tables 1 and 3.

Re: Thanks for your comments. We have modified this part. See the blue font in Table 1 and Table 3 for details.

Point 6: “The anxious arousal (AA) of the PTAD had a mean of 3.86 and an SD of 2.30”; Is PTAD a typo for PTSD in this sentence?

Re: Thank you fou reviewer′ comment. This is a mistake, we have changed “PTAD” to “PTSD”, see line 206 in blue font.

Point 7:It is written as “Overall, results suggested that the PCL scores were invariant across gender of adolescents (see Table 5)”, but there is no table 5 in the text, so please check and re-enter it.

Re: Thanks for your comments, according to experts' opinions, we have changed the wrong writing "Table 5" to "Table 4". For details, see the blue font in lines 243 and 249-250.

Point 8: Please describe the guide' questions, the guidelines for validating the model fit indexes have been describlines for validating the model fit indices below Table 4.

Re: Thanks to the expertsed in lines 161-169 and 186-197of the body. We don't think it is necessary to describe it under Table 4. If experts think it is necessary to describe it again, we will add this part to Table 4 at the next revision.

Point 9: 4. Discussion

“The αs for the PTSD was high both total scale and its subscale“; In this sentence, if you write Cronbach alpha instead of α, it will be easier for readers to recognize.

Re: Thanks for reviewer′ comment, we have changed the “αs” to “Cronbach alpha”, see the blue font in line 304 for details.

Reviewer 2 Report

The article entitled „Factor structure and psychometric properties for the PTSD checklist of Chinese adolescent in closed period after the COVID-19 outbreak” is a cognitively interesting and important study. The work presented by the authors is also of significant practical importance. Research on methods that may aid in the diagnosis and identification of people who may need psychological help in connection with the COVID-19 pandemic should be considered highly necessary. Screening methods that address the post-traumatic, psychological consequences of the SARS-CoV-2 virus pandemic seem to be particularly needed. According to numerous studies, young people, adolescents, are particularly exposed to the negative consequences of the pandemic. The study conducted by the authors should therefore be considered necessary and relevant to the group for which we will especially need diagnostic methods for psychopathological symptoms in pandemic and post-pandemic conditions. It is also worth emphasizing that the manuscript may be educational for young scientists, as it presents in an accessible and systematized way the use of various indicators and procedures useful for estimating the properties of tests.

Before publishing this article, I suggest considering a few things that could help improve the manuscript:

- Wouldn't it be more appropriate to include "Adolescents" in the title instead of Adolescent?

- In the "Participants" section, it would be good to indicate the time period in which the research was conducted.

- In the introduction, it would be worth outlining the dynamics of the pandemic, presenting basic epidemiological data on the increase in morbidity and deaths in the district where the study was conducted - it would be helpful for foreign readers in assessing the stressful situation in the district where the study was conducted.

- Were the schools in which the research was carried out randomly selected?

- In the "Instruments" section it is indicated that the PCL has a Likert scale of 0 - 5, while the minimum number of points that can be obtained is 17. How is the answer "0" converted?

- There is a repetition of one issue in the Procedure section - at the beginning and end there is the same information about "informed consent".

- In the "Analytic Plan" section, I suggest rethinking the grammatical construction used (tense). In this part, the present tense was used in relation to the description of the examination, while in the following parts the results obtained on the basis of these activities are described in the past tense, which seems not entirely logical.

- I suggest checking the work for punctuation - incl. line 52, 100, 147.

- Check the word notation on line 155 "onsistency".

- What does the abbreviation "PTAD" on line 185 stand for? Was it about PTSD?

- Check grammar on line 236 - tense used.

I hope you will find my little comments on the manuscript helpful in your work on this article.

Author Response

Dear Reviewer,

Thank you for your valuable comments! In response to your comments, we have made a detailed reply. Please check its.

Reviewer#2

Comments and Suggestions for Authors

The article entitled“Factor structure and psychometric properties for the PTSD checklist of Chinese adolescent in closed period after the COVID-19 outbreak” is a cognitively interesting and important study. The work presented by the authors is also of significant practical importance. Research on methods that may aid in the diagnosis and identification of people who may need psychological help in connection with the COVID-19 pandemic should be considered highly necessary. Screening methods that address the post-traumatic, psychological consequences of the SARS-CoV-2 virus pandemic seem to be particularly needed. According to numerous studies, young people, adolescents, are particularly exposed to the negative consequences of the pandemic. The study conducted by the authors should therefore be considered necessary and relevant to the group for which we will especially need diagnostic methods for psychopathological symptoms in pandemic and post-pandemic conditions. It is also worth emphasizing that the manuscript may be educational for young scientists, as it presents in an accessible and systematized way the use of various indicators and procedures useful for estimating the properties of tests.

Before publishing this article, I suggest considering a few things that could help improve the manuscript:

- Wouldn′t it be more appropriate to include "Adolescents" in the title instead of Adolescent?

Re: Thanks for your comment, we have modified it. See the blue font in line 3 and line 20 for details.

- In the "Participants" section, it would be good to indicate the time period in which the research was conducted.

Re: Thanks for your comment, we have explained the time period of the questionnaire in the text. See the blue font in line 118 for details.

- In the introduction, it would be worth outlining the dynamics of the pandemic, presenting basic epidemiological data on the increase in morbidity and deaths in the district where the study was conducted - it would be helpful for foreign readers in assessing the stressful situation in the district where the study was conducted.

Re: Thanks for your comment, we have added the latest data of COVID-19 patients in China and Guizhou Province to the body. See lines 24-26 of the first paragraph of the introduction in blue font fosr details. (https://voice.baidu.com/act/newpneumonia/newpneumonia/?from=osari_aladin_banner)

- Were the schools in which the research was carried out randomly selected?

Re: Thanks to the experts' questions, our participants are not completely random, but the participants are not entirely from the same region and school. Due to the epidemic prevention and control, we can only contact familiar principals in different cities of Guizhou Province for guidance and enter the school to carry out this study.

- In the "Instruments" section it is indicated that the PCL has a Likert scale of 0-5, while the minimum number of points that can be obtained is 17. How is the answer "0" converted?

Re: Thanks to the experts' comments, we have changed 0 (“never”) to 1 (“never”). See the blue font on line 133 for details.

- There is a repetition of one issue in the Procedure section - at the beginning and end there is the same information about "informed consent".

Re: Thank you for your questions. After checking, there is indeed a duplicate problem, and we have deleted it.

- In the "Analytic Plan" section, I suggest rethinking the grammatical construction used (tense). In this part, the present tense was used in relation to the description of the examination, while in the following parts the results obtained on the basis of these activities are described in the past tense, which seems not entirely logical.

Re: Thanks for the expert's questions. We have adjusted the corresponding tenses. See lines 151, 154, 156, 176, 177 and 195 in the analytical plan in blue font for details.

- I suggest checking the work for punctuation - incl. line 52, 100, 147.

Re: Thanks for your comments, we have checked and modified the punctuation of the full text.

- Check the word notation on line 155 "onsistency".

Re: Thanks for your comments, we have corrected the word, for details, see the blue font in line 172.

- What does the abbreviation "PTAD" on line 185 stand for? Was it about PTSD?

Re: Thank you for reviewer′ comment. This is a mistake, we have changed “PTAD” to “PTSD”, see line 206 in blue font.

- Check grammar on line 236 - tense used.

Re: Thanks for your comments. We have modified the tense in the paper, see the blue font in line 260 for details.

I hope you will find my little comments on the manuscript helpful in your work on this article.

Re: Thanks for your comments. Your suggestions are very helpful for us to further improve the content of the article.
